# Biocontrol of Litchi Downy Blight Dependent on *Streptomyces abikoensis* TJGA-19 Fermentation Filtrate Antagonism Competition with *Peronophythora litchii*

**Mengyu Xing [1], Dandan Xu [2], Yinggu Wu [1], Tong Liu [1], Pinggen Xi [3], Rui Wang [1], Jing Zhao [1] and Zide Jiang [3,*]**

[1] Key Laboratory of Green Prevention and Control of Tropical Diseases and Pests, Ministry of Education, School of Tropical Agriculture and Forestry, Hainan University, Haikou 570228, China; xingmengyu@hainanu.edu.cn (M.X.); wyg123452021@163.com (Y.W.); liutongamy@sina.com (T.L.); wangrui01093322@163.com (R.W.); zj90929@163.com (J.Z.)

[2] College of Life Science and Agricultural Engineering, Nanyang Normal University, Nanyang 473061, China; happyxudandan@126.com

[3] Department of Plant Pathology/Guangdong Province Key Laboratory of Microbial Signals and Disease Control, South China Agricultural University, Guangzhou 510642, China; xpg@scau.edu.cn

[*] Correspondence: zdjiang@scau.edu.cn

**Abstract:** The cultivation and overall quality of Litchi, a fruit of significant commercial value in China, are hindered by the presence of the oomycetes pathogen *Peronophythora litchii*. This pathogen is responsible for the occurrence of litchi downy blight, resulting in substantial economic losses during the storage and transportation of the fruit, and affects nutritional growth. Effective and environmentally safe methods to control litchi downy blight are urgently needed. The application of biocontrol agents such as *Streptomyces* bacteria has proven effective for controlling plant diseases. Our present study isolated the *Streptomyces* strain TJGA-19, identified as *S. abikoensis*, with potent inhibitory activity against *P. litchii*. The antifungal active substances are mainly in the aqueous phase of TJGA-19 fermentation filtrate extraction. The fermentation filtrate of TJGA-19 not only suppressed the pathogen growth, sporulation, and sporangia germination, but also delayed the disease development of litchi downy blight. In addition, the stability of the TJGA-19 fermentation filtrate was not sensitive to the proteinase K, temperature, white-flourescence light, or ultraviolet treatment. Furthermore, the morphology and ultrastructure of *P. litchii* treated with fermentation filtrate was characterized by marked shrinking and deformation, with serious disruption of plasma membrane permeabilization and the organelles. Hence, *S. abikoensis* TJGA-19 and its metabolites demonstrated marked efficiency against the phytopathogenic pathogen *P. litchii* and provide a potential candidate for controlling litchi downy blight.

**Keywords:** *Peronophythora litchii*; fermentation; litchi downy blight; *Streptomyces abikoensis*





## 1. Introduction

Litchi (*Litchi chinensis* Sonn.), a tropical to subtropical fruit, is extensively cultivated in over 20 countries globally [1]. Litchi downy blight, induced by the oomycetes pathogen *Peronophythora litchii*, is a highly destructive disease that appears to affect all litchi cultivars. This disease can infect various plant parts, including the fruit, inflorescences, tender leaves, and shoots, leading to substantial economic losses throughout the litchi pre- and post-harvest periods [2]. The application of fungicides, such as dimethomorph and azoxystrobin, is still the primary method for controlling litchi downy blight disease, even though many fungicides have been restricted due to their negative effects on human health, the environment, and pathogen resistance [3]. There is a pressing necessity to investigate alternative approaches for controlling litchi downy blight, while minimizing adverse impacts on both human health and the environment. In this context, biocontrol agents represent environmentally safer alternatives for protecting plants or fruit against disease [4].

The inherent characteristics of specific microorganisms and their metabolic processes can be manipulated for the purpose of plant disease management [5]. *Streptomyces* spp. are well-known actinomycete bacteria, which are the primary natural source of bioactive products and contribute to around 75% of all bioactive compounds, including antibiotics [6,7]. The *Streptomyces* genome contains more than 20 gene clusters to secondary metabolites that tackle the rise of antimicrobial resistance, conferring strong potential for the biocontrol of phytopathogenic microorganisms [8–10]. For instance, the strain *S. tsukiyonensis* JT-2F produce proteases and cellulase enzymes, which facilitated the degradation of protein and cellulose constituents present in the cell walls of *Colletotrichum dematium* [11], *Streptomyces* strain 135 was shown to be an excellent antimicrobial agent for controlling agricultural fungal diseases [12], and *Streptomyces* sp. strain SLR03 exhibited antifungal activity against the tea fungal plant pathogen *Pestalotiopsis theae* [13]. However, the application of antagonistic *Streptomyces abikoensis* to control litchi downy blight has not yet been addressed.

Here, we systematically test our hypothesis that *S. abikoensis* is a valuable biocontrol agent against litchi downy blight. We evaluated the potential of *S. abikoensis* as a biocontrol agent for *P. litchii* in vitro and in vivo. Moreover, we have elucidated the underlying mechanism responsible for its antimicrobial activity, thereby establishing a foundation for the potential utilization of this knowledge in the biological management of litchi downy blight.

## 2. Materials and Methods

### 2.1. Soil Samples, Pathogen, and Inoculation Materials

Rhizosphere soils of wild litchi in Bawangling Primeval Forest Reserve (Haikou City, Hainan Province, China) were collected and transferred to the laboratory for actinomycete isolation. The pathogen *P. litchii* was kindly provided by Professor Zide Jiang (South China Agricultural University, Guangzhou, China) and maintained on carrot agar (CA) media (carrot juice from 200 g of carrots, 15 g agar, water 1 L) at 25 °C. Litchi fruit (cv. Baitangying) were harvested at 80% maturation from a commercial orchard located in Haikou City, Hainan Province, China.

### 2.2. Actinomycete Isolation and Antagonistic Isolate Screen

A serial dilution method was used to isolate actinomycetes from the soil on starch casein agar (SCA) medium (starch 10 g, $K_2HPO_4$ 2 g, $KNO_3$ 2 g, casein 0.3 g, $MgSO_4 \cdot 7H_2O$ 0.05 g, $CaCO_3$ 0.02 g, $FeSO_4 \cdot 7H_2O$ 0.01 g, agar 15 g, water 1000 mL and pH 7.0 ± 0.1) [14]. After incubation at 25 °C for 5 d, actinomycete colonies with different morphological characteristics were selected and transferred to potato dextrose agar (PDA) plates, and then used for antagonistic activity testing.

The antagonistic effects of actinomycete isolates were measured using the dual culture method [15]. In brief, the colony plug (5 mm in diameter) of *P. litchii* was placed in the center of a PDA plate and four colony plugs (5 mm in diameter) of actinomycete isolates were placed at a distance of 35 mm from the pathogen plug in four different directions. After incubation for 6 d at 25 °C, the width of the inhibition zone was measured to determine the effective antagonistic activity of the actinomycete isolates.

### 2.3. Identification of Antagonistic Actinomycete TJGA-19

The actinomycete isolate TJGA-19, which demonstrated excellent inhibition activity, was inoculated on international streptomyces project (ISP) media, and the colony characteristics were observed after incubating at 28 °C for 7 d [16,17]. After 14 d, the mycelia and spores were observed under scanning electron microscopy [18].

Molecular identification of the actinomycete isolate was conducted by amplifying 16S rDNA of genomic DNA with the universal primers 27F (5′-AGAGTTTGATCCTGGCTCAG-3′) and 1492R (5′-TACGGCTACCTTGACG ACTT-3′) [19]. Polymerase chain reaction was performed with a thermal cycler (Takara) in a volume of 25 µL. The cyclic conditions were as follows: initial 3 min for denaturation at 94 °C, followed by 35 cycles of 1 min

at 94 °C, annealing at 54 °C for 1 min, extension at 72 °C for 2 min, final extension of 2 min at 72 °C, and finally, held at 4 °C. The polymerase chain reaction products were sequenced by BGI (Shenzhen, China). Subsequently, the sequences underwent a BLAST (basic local alignment search tool) search within the GenBank database, followed by the construction of a phylogenetic tree utilizing MEGA 5.1 software through the application of the neighbor-joining method [20].

### 2.4. Preparation of the TJGA-19 Fermentation Filtrate

After the antagonistic actinomycete TJGA-19 was cultured on PDA at 28 °C for 7 d, the fresh spores were collected and diluted with sterilized water to reach a concentration of $1 \times 10^8$ spores/mL. The spore suspension (0.4 mL) was mixed with 80 mL soybean meal broth medium (SBM: soluble starch 2.5%, soybean powder 5.5%, $[NH_4]_2SO_4$ 0.2%, NaCl 0.2%, yeast extract 0.1%, pH $7.0 \pm 0.1$) and incubated on a shaker at 28 °C and 200 rpm for 7 d. The resulting fermentation broth was centrifuged at $6000 \times g$ for 10 min, and then filtered through a microporous membrane (0.22 μm) to obtain the sterilized fermentation filtrate [21].

### 2.5. Measurement of the Stability of the TJGA-19 Fermentation Filtrate

The stability of the fermentation filtrate produced by TJGA-19, including its sensitivity to proteinase K, temperature, white-fluorescence light, and ultraviolet (UV) light, was assessed as previously described, with modifications [22]. Briefly, a suspension of *P. litchii* sporangia (100 μL) was added to solidified PDA medium and evenly spread using a coating rod. A well was created in the center of the PDA plates using a sterile punch ($\varphi$ = 5 mm), and the agar block was removed. Subsequently, 100 μL of sterile fermentation filtrate was added to the well. The plates were then incubated at 25 °C for 6 days, and the diameter of the inhibition zone was measured using a cross method to determined antimicrobial activity. All experiments were performed in triplicate and three replicates for each treatment.

### 2.6. Extraction of Bioactive Metabolites and Determination of Antifungal Activity

A 2 L fermentation broth of TJGA-19 underwent extraction using petroleum ether and ethyl acetate solvents at ambient temperature. Each solvent was subjected to three extraction cycles, and the resulting solvent liquids were combined and concentrated under reduced pressure to yield petroleum ether extracts (PEEs), ethyl acetate extracts (EAEs), and aqueous phase extracts (APEs), which were subsequently dried using a rotary evaporator [13]. The PEEs and EAEs were dissolved in dimethyl sulfoxide (DMSO), while the APEs were dissolved in sterile water. The antimicrobial activity was assessed following the methodology outlined in Section 2.5. A 100 μL solution of PEEs, EAEs, and APEs was placed in the well; the concentration of each extract was 50 mg/mL, with SBM extracts (SBMs) and DMSO used as a negative control at the same concentration. All experiments were performed in triplicate.

### 2.7. Bioactivity Assay on the TJGA-19 Fermentation Filtrate

The efficacy of the TJGA-19 fermentation filtrate in suppressing the mycelial growth of *P. litchii* was assessed by incorporating the fermentation filtrate into PDA medium and subsequently adjusting it to the desired concentration. (0.25%, 0.5%, 1%, 2%, and 4%). An equal volume of SBM without fermentation filtrate added into the PDA medium was treated as a control. Then, plug inocula (diameter of 6 mm) were cut from an active *P. litchii* colony and inoculated onto the center of the plate. Colony diameters and mycelial growth inhibition were measured after incubation at 25 °C for 7 days. Treatment at each concentration consisted of four replicates, and this experiment was conducted twice.

To test the effect of TJGA-19 fermentation filtrate on the germination of *P. litchii* sporangia, the porangias suspension was mixed with TJGA-19 fermentation filtrate to achieve final concentrations of 0.25%, 0.5%, 1%, 2%, 3%, 5%, and 10%, respectively. An equal volume of SBM without fermentation filtrate was used as a control. After all sporangia

suspensions were incubated at 25 °C for 2 h, the germinated sporangia were measured, and the sporangial germination inhibition was calculated. Each treatment concentration was replicated four times independently, and the experiment was repeated three times.

### 2.8. Plasma Membrane Permeabilization Assay

The plasma membrane permeabilization of *P. litchii* was assessed using a Sytox green uptake assay, as described by Taveira et al. [23]. To obtain fresh mycelia, a suspension of *P. litchii* sporangia ($1 \times 10^5$ sporangia/mL) was incubated in potato dextrose broth at 25 °C and 180 rpm for 36 h. The mycelia were harvested and fermentation filtrate was added to adjust to final concentrations of 2% and 5%. Mycelia treated with SBM without fermentation filtrate served as a control. After 6 h, 0.8 μM Sytox green was added and incubated for 30 min in the dark. The stained mycelia underwent three rounds of washing with phosphate-buffered saline and were subsequently observed using an optical microscope (DMi8, Leica, Wetzlar, Germany) equipped with a specialized fluorescent filter set optimized for the identification of fluorescein, featuring excitation wavelengths of 450 nm and emission wavelengths of 540 nm.

### 2.9. The Effect of Fermentation Filtrate on the Morphology and Ultrastructure of P. litchii

TJGA-19 fermentation filtrate was added into PDA medium to reach the final concentration of 2%, and then the *P. litchii* was inoculated and incubated at 25 °C for 6 d. An equal volume of SBM without fermentation filtrate was added as a control. Mycelia and sporangia were collected and prepared for SEM and transmission electron microscopy (TEM) observations [24]. The samples underwent analysis using an LEO-1530VP scanning electron microscope (LEO, Oberkochen, Germany) at an operating voltage of 5 kV, and a transmission electron microscope (Tecnai, FEI, Hillsboro, OR, USA) at an operating voltage of 100 kV, for the purpose of conducting SEM and TEM analyses, respectively.

### 2.10. In Vivo Bioassay on Detached Litchi Leaf and Litchi Fruit

Fifteen litchi tender leaves (upside-down) were arranged in a circular pattern on a Petri dish with a diameter of 20 cm, which was lined with moist sterile filter paper to maintain the desired level of humidity. A volume of 2 μL of sporangia suspension, containing $2 \times 10^4$ sporangia/mL, was dipped to the back of each leaf blade. Subsequently, different concentrations (2%, 5%, 10%, and 20%) of TJGA-19 fermentation filtrate were sprayed onto the leaves, while an equal volume of SBM was used as a control. Following an incubation period of 48 h at a temperature of 25 °C, the incidence rate and lesion length were determined. Each treatment involved 45 leaves, and the experiment was replicated three times.

Litchi fruits with uniform size and maturity were selected and washed with running water twice, then air-dried at 25 °C for 30 min. All litchi fruits were inoculated with *P. litchii* by dipping 10 μL sporangia suspension ($2 \times 10^4$ sporangia/mL) and sprayed TJGA-19 fermentation filtrate, the concentrations of fermentation filtrate were adjusted from 2% to 20% and equal volume of SBM served as a control. For each treatment concentration, 75 fruits were treated and three replicates were conducted. All treated litchi fruit were kept in plastic containers and incubated at 25 °C, and disease development on the fruit was observed each day.

### 2.11. Statistical Analysis

The data were analyzed using SPSS 23 statistical software (SPSS, Chicago, IL, USA) for the purpose of conducting an analysis of variance (ANOVA). To compare the various treatments, Duncan's multiple range test was utilized at a confidence level of 95% ($p \leq 0.05$).

## 3. Results

### 3.1. Isolation, Screening Antagonistic Actinomycetes, and Identification of TJGA-19

In total, 476 actinomycetes were obtained from 28 soil samples, and 163 representative strains with different characteristics were selected for further experiment (Figure S1). According to the antagonistic test, 27 actinomycetes exhibited strong inhibitory activity against *P. litchii*. Among these, strain TJGA-19 showed the strongest inhibitory activity (width of inhibitory zone: 21.4 mm [Table S1]) and was selected for further study. The strain TIGA-19 was identified as *S. abikoensis* by morphology characteristics, physiological methods and molecular identification (Figure S2).

### 3.2. Determination of the Stability of TJGA-19 Fermentation Filtrate

To assess the stability of the TJGA-19 fermentation filtrate, physical factors including temperature and UV light were applied and the mycelial growth inhibition rate was used as the indicator of stability. Our result indicated that proteinase K treatment did not affect inhibition activity (Figure 1A). Furthermore, the growth of *P. litchii* mycelium was slightly affected under the treatment of white-flourescence light, ultraviolet light, temperature (28–121 °C) and temperature of 100 °C for 0–120 min, while without significant difference (Figure 1B–E). From these results, we inferred that the fermentation filtrate of strain TJGA-19 showed good physical stability.

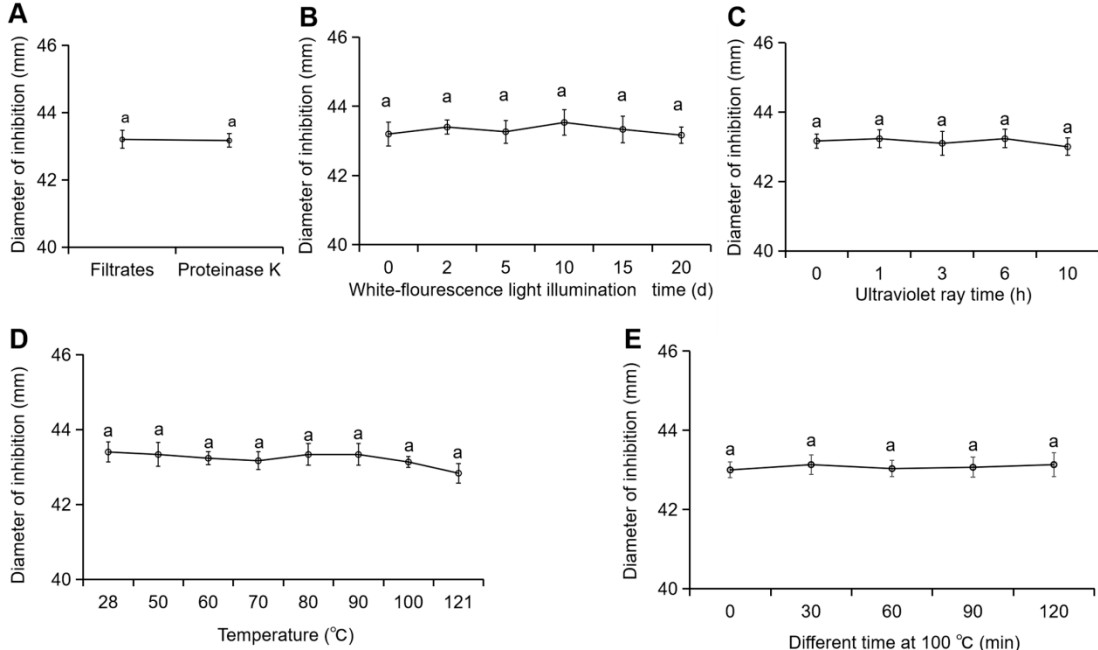

**Figure 1.** Stability of the TJGA-19 fermentation filtrate. (**A**): proteinase K treatment; (**B**): white flourescence light; (**C**): ultraviolet light; (**D**): temperature; (**E**): temperature of 100 °C for 0–120 min. Data presented are means ± SE (*n* = 9), and same letter indicate no significantly different. ($p \leq 0.05$).

### 3.3. Antifungal Activity of the TJGA-19 Fermentation Filtrate Extraction

Since actinomycete strain TJGA-19 showed the strongest inhibitory activity against *P. litchii*, we inferred that its fermentation filtrate extraction might have attributed to the suppression. Hence, we tested the effects of different extraction phases of TJGA-19 fermentation filtrate on the mycelial growth. The findings indicated that APEs exhibited the highest level of inhibition on the growth of *P. litchii* colonies, the inhibition zone diameter was 7.2 mm, while SBMs, DMSO, PEEs and EAEs showed no antifungal activity (Figure 2). This indicates that the antifungal active substances are mainly in the aqueous phase, so the fermentation broth is used for further experiments.

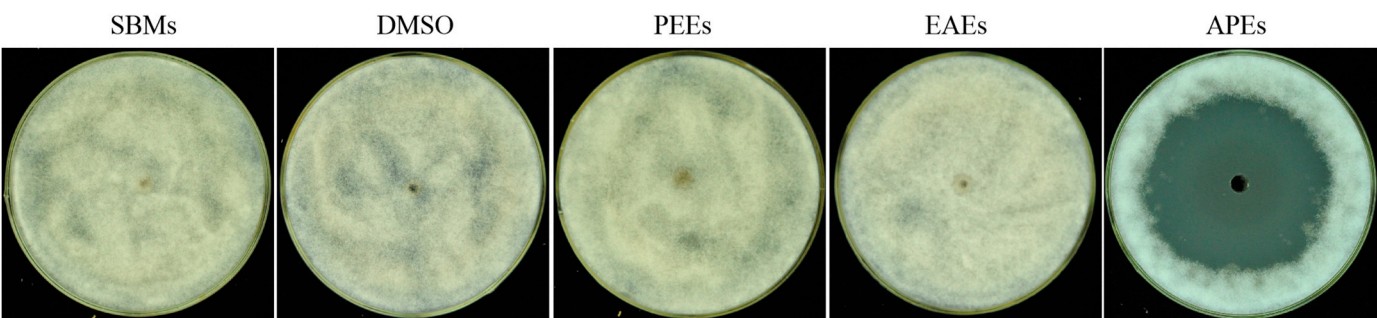

**Figure 2.** Antifungal activity of the various extraction phases of TJGA-19 fermentation filtrate against *P. litchii*.

### 3.4. Inhibitory Effect of TJGA-19 Fermentation Filtrate

In the above assay, APEs showed the strongest inhibition effect against *P. litchii*, which indicated that TJGA-19 fermentation filtrate can have the greatest antifungal effect. Hence, we tested the effects of different concentrations of TJGA-19 fermentation filtrate on the growth of mycelia, sporulation, and sporangia germination of *P. litchii*. As shown in Figure 3A,B, fermentation filtrate at concentration ranging from 0.25% to 4% significantly inhibited the mycelial growth of *P. litchii*, with total inhibition when the concentration reached 4%. Sporangia germination was also markedly suppressed by treatment with fermentation filtrate compared to control treatment, again with a concentration response. Sporangia germination was totally inhibited at a concentration of 4% (Figure 3C).

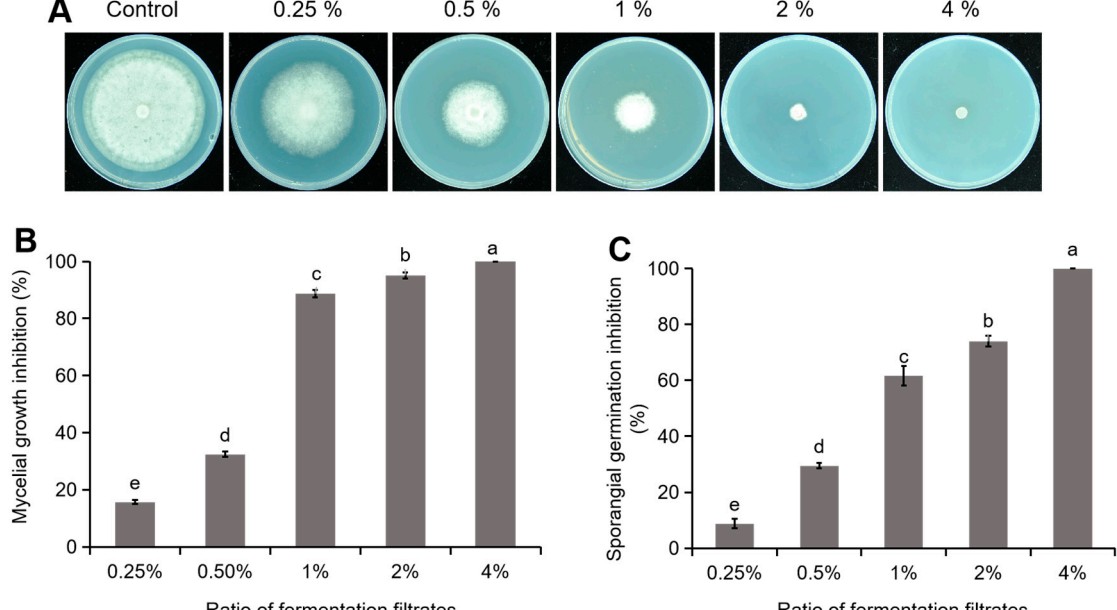

**Figure 3.** Suppression activity of TJGA-19 fermentation filtrate against *P. litchii*. (**A**): mycelial growth of *P. litchii* was observed after incubating on PDA medium with different concentrations of fermentation filtrate for 7 d; (**B**): the inhibitory activity was determined, (**C**): Investigate the germination of sporangia in *P. litchii* after a suspension of sporangia represent subjected to various concentrations of fermentation filtrate for 2 h. The presented data represent the means ± standard error. In the accompanying graphs, lowercase letters above the bars indicate significant differences as determined by statistical analysis using SPSS 23 with Duncan's multiple range test ($p \leq 0.05$).

### 3.5. Disruption of Plasma Membrane Permeabilization of P. litchii

To determine the action mechanism of the TJGA-19 fermentation filtrate, we tested membrane permeabilization using Sytox green staining. Mycelia and sporangia in the

control group showed weak fluorescence, while those treated with fermentation filtrate at concentrations of 2% and 5% showed strong fluorescence (Figure 4). This experiment indicated that application of the TJGA-19 fermentation filtrate resulted in impaired plasma membrane permeabilization in *P. litchii*.

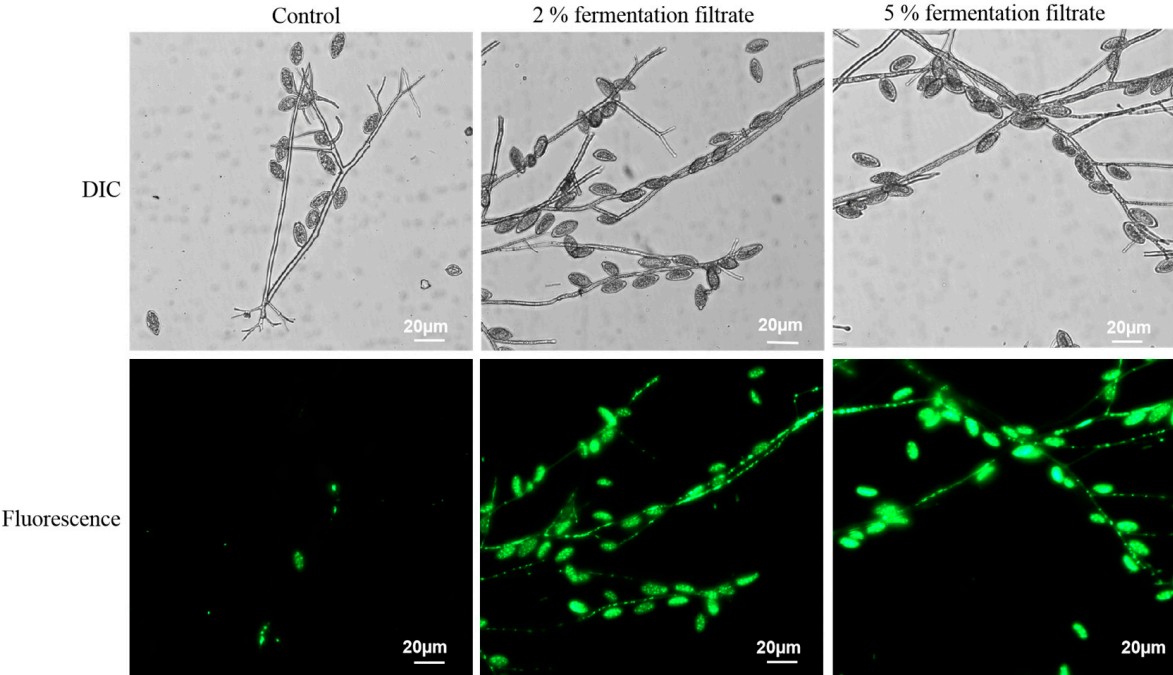

**Figure 4.** The effect of *S. abikoensis* TJGA-19 fermentation filtrate on the permeabilization of the membrane in *P. litchii* was evaluated through the utilization of a Sytox green uptake assay. The pathogen was incubated for 6 h with SBM without fermentation filtrate as control treatment, 2% TJGA-19 fermentation filtrate, and 5% fermentation filtrate, then the fluorescein were detected.

*3.6. Alteration of the Morphology and Ultrastructure of P. litchii*

To check for changes in pathogen morphology and ultrastructure caused by TJGA-19 fermentation filtrate, we conducted SEM and TEM observations. SEM observation of mycelia in the control treatment group were smooth, straight, regular, and uniform and the sporangia were normal, oval, and plum (Figure 5A,B). Meanwhile, mycelia treated with TJGA-19 fermentation filtrate were severely shrunken, collapsed, and deformed, and no sporangia formed (Figure 5C,D; arrow shown). TEM analysis revealed that the hyphae in the control group exhibited typical morphology in the cell wall, plasma membrane, cytoplasm, vacuoles, and mitochondria, as depicted in Figure 6A,B. Conversely, the group subjected to TJGA-19 fermentation filtrate treatment exhibited significant damage or complete destruction of vacuoles and organelles. (Figure 6C,D; arrow shown). This work showed that the effects of TJGA-19 fermentation filtrate on the *P. litchii* pathogen were strongly dependent on damage to the membranes and organelles.

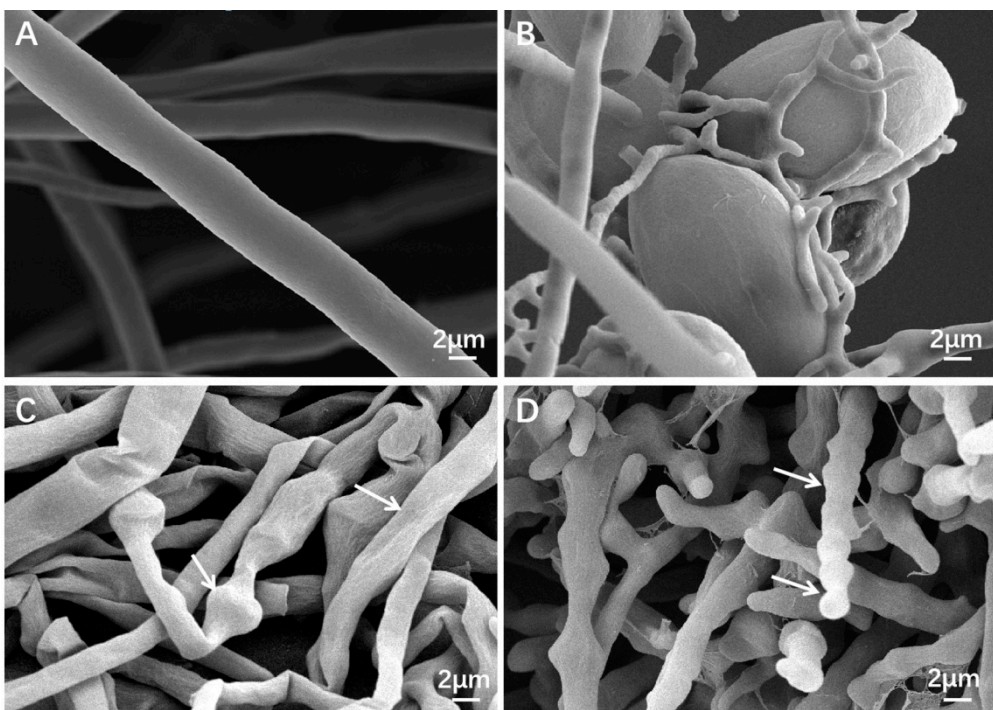

**Figure 5.** *S. abikoensis* TJGA-19 fermentation filtrate caused morphological changes in *P. litchii* hyphae and sporangia. (**A**,**B**): untreated control; (**C**,**D**): mycelia were inoculated on carrot agar media with 2% *S. abikoensis* TJGA-19 fermentation filtrate for 6 d. Arrows denote shrinking or distorted mycelia.

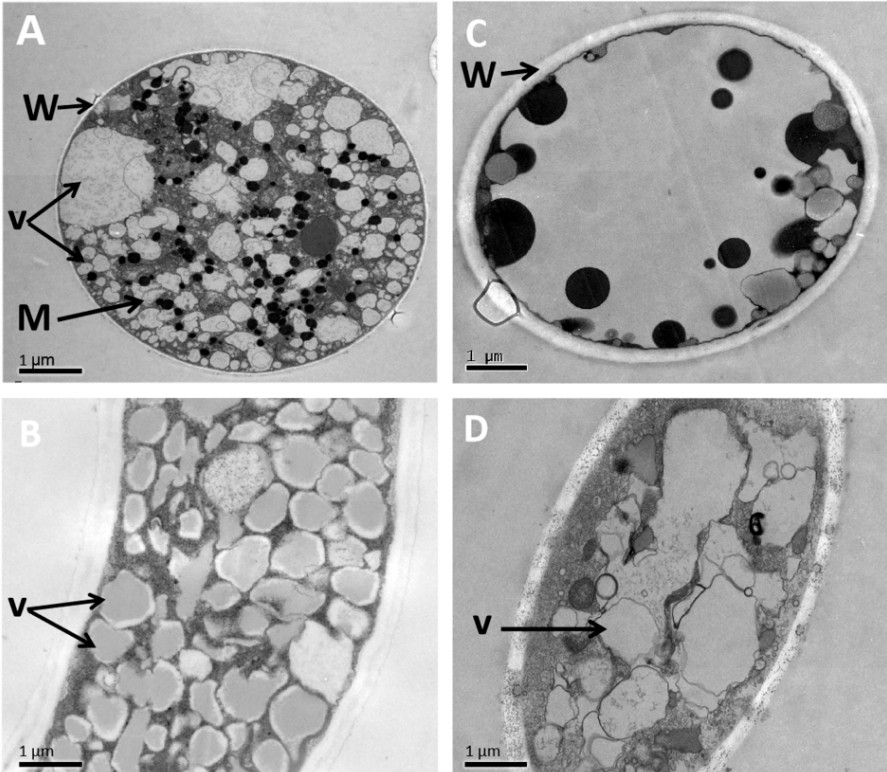

**Figure 6.** Pathogen cellular damage caused by *S. abikoensis* TJGA-19 fermentation filtrate. (**A**,**B**): control treatment. (**C**,**D**): the pathogen was treated with 2% *S. abikoensis* TJGA-19 fermentation filtrate for 6 d. A and C, tangential section through the hyphae of *P. litchii*; B and D, longitudinal section through the hyphae of *P. litchii*. M, mitochondria; V, vacuoles; W, cell wall.

### 3.7. Delayed Disease Development of Litchi Downy Blight

To measure the efficacy of TJGA-19 fermentation filtrate on the control of litchi downy blight, litchi leaf and fruit were inoculated with *P. litchii*, then sprayed with fermentation filtrate, and then observed for disease development. The results shown that fermentation filtrate application effectively delayed the disease development. Inoculation results on litchi leaves showed that the control group exhibited an incidence rate of lithci downy blight at 97.3%, with a lesion diameter measuring 18.3 mm. Conversely, when the concentration of fermentation filtrate was adjusted to 2%, 5%, and 10%, the incidence rates decreased to 76.5%, 38.9%, and 24.1%, respectively. These rates were significantly lower than those observed in the control group. Additionally, the lesion diameter decreased from 7.5 mm to 2.6 mm (Figure 7A–C).

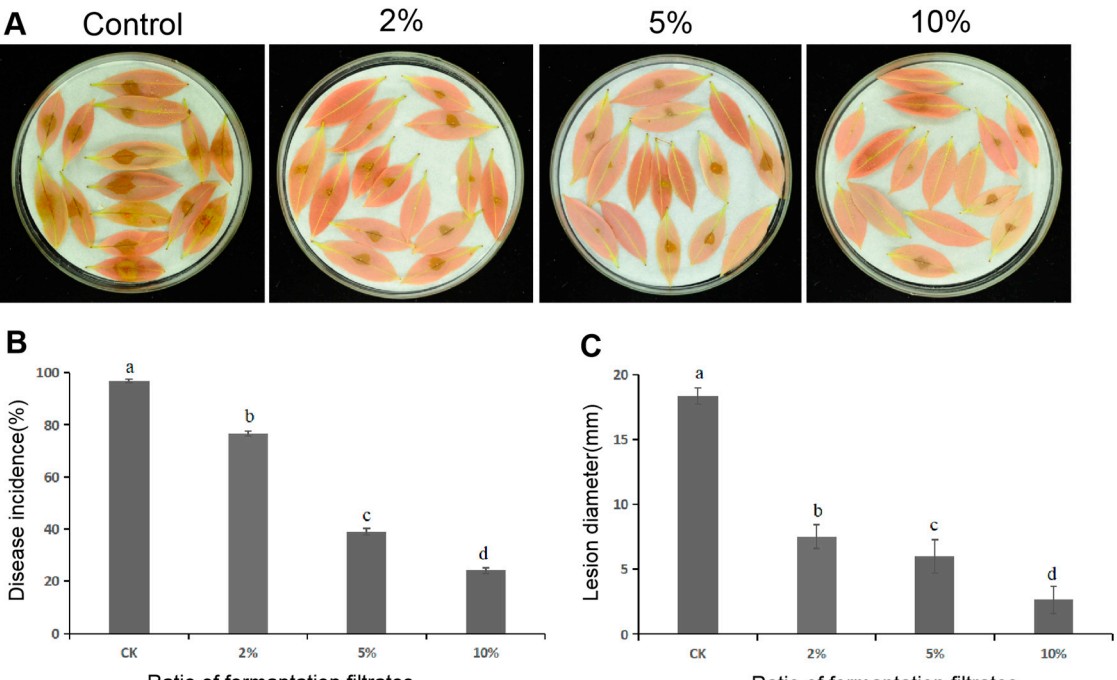

**Figure 7.** Antifungal activity of TJGA-19 fermentation filtrate against *P. litchii* on detached leaves. (**A**): the areas of leaf lesions were observed at 48 h after inoculation with *P. litchii* in both the control group and the group treated with varying concentrations of TJGA-19 fermentation filtrate. (**B,C**): at 48 h post inoculation, disease incidence and lesion length were measured. The mean ± standard error was calculated based on three replicates. Values that were denoted by different letters were found to be significantly different ($p \leq 0.05$).

At 72 h post inoculation, the fruits exhibited prominent dark-brown lesions accompanied by white sporangiophores in the control group, while only minimal browning was observed on litchi fruits treated with fermentation filtrate at a concentration of 20% (Figure 8A). The incidence of litchi downy blight in fruit treated with fermentation filtrate (20%) was reduced to 25.3%, which was significantly lower than that of the control treatment 93.7% (Figure 8B). In addition, TJGA-19 fermentation filtrate exhibited excellent efficacy for delaying browning: the browning index after treatment with 5%, 10%, and 20% fermentation filtrate was significantly lower than that in the control group (Figure 8C). This result indicates that 20% fermentation filtrate effectively suppressed the development of downy blight in litchi leaf and fruit.

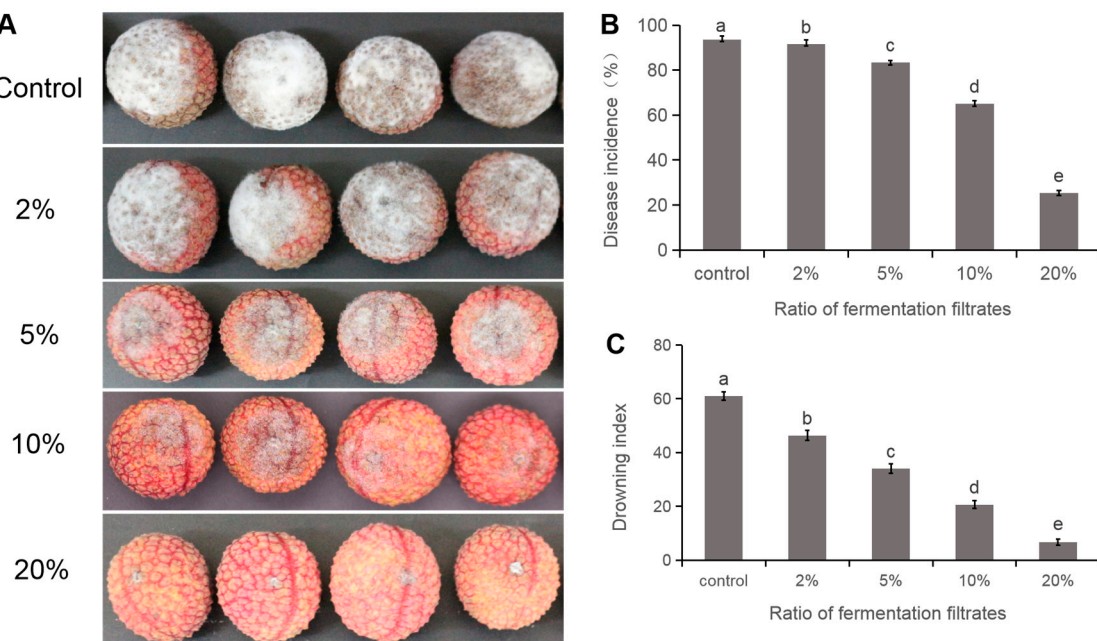

**Figure 8.** Efficacy of *S. abikoensis* TJGA-19 fermentation filtrate for suppressing litchi downy blight disease development on postharvest litchi fruit. (**A**): disease symptoms were observed after incubation at 25 °C for 72 h; (**B**): disease incidence; (**C**): browning index. Data presented are means ± SE. In the graphs, lowercase letters above the bars indicate significant differences according to statistical analysis using SPSS 23 with Duncan's multiple range test ($p \leq 0.05$).

## 4. Discussion

The extensive research on the *Streptomyces*-mediated biocontrol of plant disease is attributed to the remarkable efficacy of *Streptomyces* in synthesizing functional metabolites that possess the ability to eradicate or impede the growth of plant pathogens [25–27]. Notably, desertomycin, spectinomycin, nigericin, and validamycin, which are bioactive metabolites derived from *Streptomyces*, have exhibited potent antimicrobial properties [28,29]. *Streptomyces* spp. have been identified as promising biocontrol agents for the management of various plant diseases. For example, *Streptomyces* isolates MBFA-172 and H4 reduced postharvest anthracnose disease of the strawberry [30,31]; *Streptomyces* sp. HSL-9B, isolated from mangrove forest exhibited potent antifungal activity against *Colletotrichum gloeosporioides* and could decrease mango decay during postharvest storage [32,33]; and *S. violascens* MT7 showed strong potential to reduce sour-rot development in citrus and soft-rot development in papaya fruits [33]. However, few reports focused on the application of *Streptomyces* in controlling litchi downy blight pathogen. Our research will enrich this field.

The accumulated evidence demonstrates that the fermentation filtrate of *Streptomyces* spp. contributes to its inhibitory activity against the growth of pathogenic mycelia and conidia [34]. The fermentation filtrate derived from *Streptomyces* sp. JCK-6131 demonstrated effective protection against bacterial and fungal pathogens by exhibiting a broad-spectrum antimicrobial activity and inducing plant systemic resistance [35]. Additionally, Evangelista-Martínez et al. [36] found that the bioactive extract obtained from the novel *Streptomyces* strain CACIS-1.5CA exhibited strong inhibition of spore germination in postharvest fruit pathogen, including *Colletotrichum*, *Alternaria*, *Aspergillus*, *Botrytis*, *Rhizoctonia*, and *Rhizopus*. Our present results indicate that the fermentation filtrate of *Streptomyces* TJGA-19 is a strong inhibitor of *P. litchii*, this suggests the fermentation filtrate could be used as a promising alternative for preserving litchi freshness during the postharvest stage. In addition, our observation that proteinase K application did not affect the antagonistic activity of the *Streptomyces* TJGA-19 fermentation filtrate highlights that its high stability depends on non-enzymatic activity and promotes its application for postharvest disease control.

To uncover the mechanism contributing to the suppression of *P. litchii* plasma membrane permeabilization, we analyzed the mycelial and sporangial morphology and ultrastructure of *P. litchii* after treatment with *Streptomyces* TJGA-19 fermentation filtrate. Like other antimicrobial compounds, the fermentation filtrate disrupted the *P. litchii* plasma membrane and induced mycelial malformation and cellular melt, thus leading to cell death. In our previous study, similar morphological and ultrastructural abnormalities of *P. litchii* were observed after treatment with pterostilbene [27]. The degradation of organelles and nucleus is thus the major mechanism by which the *Streptomyces* TJGA-19 fermentation filtrate inhibits pathogen growth. We speculated that *Streptomyces* TJGA-19 might produce metabolites that improve its antagonistic activity. Hence, future studies will be needed to identify these key compounds.

## 5. Conclusions

In our current investigation, the newly isolated *Streptomyces* TJGA-19 exhibited robust antagonistic properties against *P. litchii.* The TJGA-19 fermentation filtrate significantly inhibited the mycelial growth and sporangial germination of *P. litchii* and delayed the development of litchi downy blight throughout its growth, development, and postharvest storage. Based on stability tests under exposure to proteinase K and a range of temperature, white-flourescence light, and UV light levels, the *Streptomyces* TJGA-19 fermentation filtrate was found to be stable. Obvious disruption of the plasma membrane, mycelial malformation, and cellular melt were detected in *P. litchii* treated with fermentation filtrate, which may be due to the antimicrobial activity of *Streptomyces* TJGA-19. To our knowledge, this is the first report exploring the potential of *Streptomyces* TJGA-19 as a biocontrol agent for controlling litchi downy blight.

**Supplementary Materials:** The following supporting information can be downloaded at: https://www.mdpi.com/article/10.3390/fermentation9121011/s1, Figure S1: The colony morphology of some actinomycetes strains; Figure S2: Cultural and morphological characteristics and phylogenetic tree based on 16S rDNA sequences of antagonistic actinomycete strain TJGA-19. Table S1: The inhibition of actinomycetes against *P. litchii.*

**Author Contributions:** M.X. and D.X.: Writing original draft; Y.W.: methodology; P.X.: data curation and Investigation; T.L. and R.W.: methodology; J.Z.: investigation; Z.J.: designed the experiments and review. All authors have read and agreed to the published version of the manuscript.

**Funding:** This research was supported by the earmarked fund for Key-Area Research and Development Program of Guangdong Province (2018B020205003), the China Agriculture Research System (CARS-32), and the National Natural Science Foundation of China (Project No. 32260649), and the Natural Science Foundation of Hainan Province (320RC482, 323RC405), and the project "Basic substances as an environmentally friendly alternative to synthetic pesticides for plant protection (BasicS) (2020-C-353)".

**Institutional Review Board Statement:** Not applicable.

**Informed Consent Statement:** Not applicable.

**Data Availability Statement:** The authors of this article will provide the raw data that support the conclusions, without any unwarranted hesitation.

**Acknowledgments:** We are grateful to Norvienyeku Justice of School tropical Agriculture and Forestry, Hainan University for critical reading and useful advice on revision of the manuscript.

**Conflicts of Interest:** The authors declare no conflict of interest.

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
