# Peer review of "Biocontrol of Litchi Downy Blight Dependent on Streptomyces abikoensis TJGA-19 Fermentation Filtrate Antagonism Competition with Peronophythora litchii"

_fermentation, doi:10.3390/fermentation9121011_

Round 1

Reviewer 1 Report

Comments and Suggestions for Authors

Dear Editor and Authors,

Xing et al. carried out significant work demonstrating the potential of an actinomycete strain in controlling a critical fungal disease that affects lychee crops. The study is complete in demonstrating through several experiments how the pathogen is affected by actinomycete extracts: paired tests in Petri dishes, effects on leaves and fruits, and excellent electron microscopy photos. Regarding identifying the actinomycete, I would like to know if this was done in another study and if it would be unnecessary to repeat the entire methodology used on that occasion. Is this a new study?

The manuscript needs a complete spelling check for better understanding.

Specific concerns

Numbered lines make review work easier.

1st paragraph - Litchi chinensis in italics.

2nd paragraph -  L6 - Streptomyces tsukiyonensis in italics.

3rd paragraph – L3 – in vivo and in vitro, in italics.

Material and Methods

1st paragraph -L3 – was was.

2.2. Actinomycete isolation and antagonistic isolate screen – Please provide the pH of each medium.

2.3. Identification of antagonistic actinomycete TJGA-19 – L3 – the mycelium; L4 – observed instead observation.

L160 – "equipped".

L197-201 - Are these results from another study that has already been published? If so, would not the initial mention of the origin of this actinomycete be enough?

Figure 1 should be presented in the microscopy images and the phylogenetic tree, with more information about the methods used: Fig. 1 and Fig. 2.

Figure 3 A - The image should be better aligned with the caption.

Figure 5: The image should be better aligned with the caption.

Comments on the Quality of English Language

Dear Editor and Authors,

Xing et al. carried out significant work demonstrating the potential of an actinomycete strain in controlling a critical fungal disease that affects lychee crops. The study is complete in demonstrating through several experiments how the pathogen is affected by actinomycete extracts: paired tests in Petri dishes, effects on leaves and fruits, and excellent electron microscopy photos. Regarding identifying the actinomycete, I would like to know if this was done in another study and if it would be unnecessary to repeat the entire methodology used on that occasion. Is this a new study?

The manuscript needs a complete spelling check for better understanding.

Specific concerns

Numbered lines make review work easier.

1st paragraph - Litchi chinensis in italics.

2nd paragraph -  L6 - Streptomyces tsukiyonensis in italics.

3rd paragraph – L3 – in vivo and in vitro, in italics.

Material and Methods

1st paragraph -L3 – was was.

2.2. Actinomycete isolation and antagonistic isolate screen – Please provide the pH of each medium.

2.3. Identification of antagonistic actinomycete TJGA-19 – L3 – the mycelium; L4 – observed instead observation.

L160 – "equipped".

L197-201 - Are these results from another study that has already been published? If so, would not the initial mention of the origin of this actinomycete be enough?

Figure 1 should be presented in the microscopy images and the phylogenetic tree, with more information about the methods used: Fig. 1 and Fig. 2.

Figure 3 A - The image should be better aligned with the caption.

Figure 5: The image should be better aligned with the caption.

Reviewer 2 Report

Comments and Suggestions for Authors

Title of the point 2.5. "Measurement of the stability of the TJGA-19 fermentation filtrate" is too some extent inadequate.

In fact you have also presented a method of antimicrobial activity of the strain TJGA-19, which is not mentioned in the title of this point. 

Although I am not qualified to assess the quality of English, my impression is that English should be improved. I see gramatical and typing errors. Better English will improve overall quality of the text. 

Reviewer 3 Report

Comments and Suggestions for Authors

The publication contains very interesting study with high scietific level. the methodology is described perfectly. The results are presented clearly. the conclusions are formulated very clearly. Some spelling mistakes I mentioned in attached pdf file.

I have only one question to authors:

Are you going to patent and commerciate the S. abikoensis TJGA-19 ?
